# A Second Wind for Inorganic APIs: Leishmanicidal and Antileukemic Activity of Hydrated Bismuth Oxide Nanoparticles

**DOI:** 10.3390/pharmaceutics16070874

**Published:** 2024-06-29

**Authors:** Andriy Grafov, Ana Flávia da Silva Chagas, Alice de Freitas Gomes, Wessal Ouedrhiri, Pierfrancesco Cerruti, Maria Cristina Del Barone, Breno de Souza Mota, Carlos Eduardo de Castro Alves, Anny Maíza Vargas Brasil, Antonia Maria Ramos Franco Pereira, Gemilson Soares Pontes

**Affiliations:** 1Department of Chemistry, University of Helsinki, A.I. Virtasen Aukio 1 (PL 55), 00560 Helsinki, Finland; 2Multi-User Center for Analysis of Biomedical Phenomena, State University of Amazonas, Manaus 69065-001, AM, Brazil; 3Post-Graduate Program in Hematology, The State University of Amazon, Foundation of Hematology and Hemotherapy of Amazonas, Manaus 69050-010, AM, Brazil; 4Laboratory of Virology and Immunology, INPA, Manaus 69067-375, AM, Brazil; 5Institute for Polymers, Composites, and Biomaterials, National Research Council, 80078 Pozzuoli, NA, Italy; 6School of Nursing, University of São Paulo, São Paulo 05403-000, SP, Brazil; 7Coordination of Biomedicine, FAMETRO University Center, Manaus 69050-000, AM, Brazil; 8Laboratory of Leishmaniasis and Chagas Disease, National Institute of Amazonian Research (INPA), Manaus 69067-375, AM, Brazil

**Keywords:** leishmaniasis, nanoparticles, bismuth, anticancer, myeloid leukemia

## Abstract

American cutaneous leishmaniasis is a disease caused by protozoa of the genus Leishmania. Currently, meglumine antimoniate is the first-choice treatment for the disease. The limited efficacy and high toxicity of the drug results in the necessity to search for new active principles. Nanotechnology is gaining importance in the field, since it can provide better efficacy and lower toxicity of the drugs. The present study aimed to synthesize, characterize, and evaluate the in vitro leishmanicidal and antileukemic activity of bismuth nanoparticles (BiNPs). Promastigotes and amastigotes of *L.* (*V.*) *guyanensis* and *L.* (*L.*) *amazonensis* were exposed to BiNPs. The efficacy of the nanoparticles was determined by measurement of the parasite viability and the percentage of infected cells, while the cytotoxicity was characterized by the colorimetry. BiNPs did not induce cytotoxicity in murine peritoneal macrophages and showed better efficacy in inhibiting promastigotes (IC50 < 0.46 nM) and amastigotes of *L.* (*L.*) *amazonensis*. This is the first report on the leishmanicidal activity of Bi-based materials against *L.* (*V.*) *guayanensis*. BiNPs demonstrated significant cytotoxic activity against K562 and HL60 cells at all evaluated concentrations. While the nanoparticles also showed some cytotoxicity towards non-cancerous Vero cells, the effect was much lower compared to that on cancer cells. Treatment with BiNPs also had a significant effect on inhibiting and reducing colony formation in HL60 cells. These results indicate that bismuth nanoparticles have the potential for an inhibitory effect on the clonal expansion of cancer cells.

## 1. Introduction

Bismuth is one of the ten metals known from ancient times, but still, one of the least understood elements of the periodic system. It is the least toxic heavy metal that has been considered as the heaviest stable metal throughout the history until a negligible radioactive decay of a primordial ^209^Bi isotope was discovered in 2003 [1]. Nevertheless, in terms of chemical and pharmaceutical properties, we can still consider it stable, since the half-life of the ^209^Bi is more than billion times longer than the time lapsed from the Big Bang of the universe (*sic!*). Curative properties of bismuth compounds were first discovered in the Middle Ages to treat dyspepsia [2]. Since that time, the area of their therapeutic application has become much broader and includes the treatment of gastrointestinal disorders, syphilis, hypertension, and different infections as well as skin care and cosmetics [3,4,5]. Several compounds of bismuth, such as subcitrate (De-Nol), subsalicilate (Pepto-Bismol), subgallate, subnitrate, tartrate, and ranitidine citrate (Pylorid) are among the most widely known antimicrobials in the world [5,6,7,8,9]. Particularly, bismuth salts are recognized for their strong antibacterial action against *Helicobacter pylori* [9]. A considerable deal of research on Bi compounds has been targeted at the development of new contrast agents for imaging, antiviral and anti-inflammatory drugs, and antimicrobial metallopharmaceuticals with antitumor and leishmanicidal activity [6,10,11,12], since clinical aspects of leishmaniasis sometimes closely resemble those observed in various neoplasms [13].

Leishmaniasis is a neglected parasitic disease caused by protozoa of the genus *Leishmania*, which can provoke a cutaneous or visceral form of the disease in a mammalian host [14,15]. Worldwide, an estimated 12 million people are infected with *Leishmania* spp., and additional 350 million are at risk of infection [16]. In 2022 alone, more than 218,000 new cases of the disease were reported [17].

Current chemotherapy for leishmaniasis is based mostly on century-old pentavalent antimonials [18], such as currently commercialized intravenous (IV) sodium stilbogluconate (Pentostam^®^) or intramuscular (IM) meglumine antimoniate (Glucantime^®^); IV amphotericin B or liposomal amphotericin B; IM pentamidine isethionate; and oral miltefosine [19]. The medication choice and the administration route depend on the clinical manifestations, the species involved, the response to the treatment, and the availability of the medicines [14,20]. However, the described therapeutic scheme has several limitations related to the low efficacy and high toxicity of the drugs, long and painful treatment, development of high resistance in the parasites, and elevated costs. Therefore, there is an urgent need for new active ingredients that would effectively treat the disease [21,22]. 

From that viewpoint, Bi is an interesting and promising alternative, since on the one hand it is a close antimony analogue, and on the other hand it is a biologically active metal possessing a low toxicity to mammalian cells [19,23]. Up to date, the leishmanicidal activity of bismuth compounds has been studied for metal complexes only [6,19]; to the best of our knowledge, this is the first report on Bi-containing nanoparticles (NPs) evaluation for antileishmanial activity. 

Leukemias are a group of malignant neoplasms, originating from hematopoietic stem cells, characterized by the proliferation and accumulation of leukocytes in the bone marrow and blood [24]. As of 2018 (the most recent data available), there were 249,000 new cases of leukemia worldwide [25] representing the 11th most common cancer in the global context [25]. e.g., in Brazil, there were 11,540 new cases of leukemia for each year of the 2021–2023 triennium according to estimations of the National Cancer Institute (INCA), and 6250 cases of those happened in men and 5290 in women [26].

Treatment for leukemia includes chemotherapy, radiotherapy, and bone marrow transplantation, which can cause several long-lasting side effects [27,28]. Multidrug resistance (MDR) represents a major challenge for the effectiveness of chemotherapy, especially in hematological malignancies [29]. Predominantly, the resistance of the neoplastic cells to the cytotoxic actions of chemotherapeutic agents appears to be due to an increased efflux of those drugs through negative regulation pathways and to the excretion process mediated by P-glycoprotein [29]. 

Regarding the antitumor activity, a wide variety of bismuth complexes and organobismuth compounds demonstrated promising anticancer properties against different cancer cell lines [6,30]. Nanotechnology is a promising therapeutic strategy for the more targeted delivery of active compounds to the target cells, providing high specificity and selectivity and thus reducing toxicity [31,32]. Nanoparticles have been mostly used as drug carriers to increase the effectiveness of chemotherapy drugs by protecting the active compounds from degradation and improving their absorption, bioavailability, and solubility [33]. A promising pharmaceutical potential was also reported for lipophilic bismuth nanoparticles treated with 2,3-dimercapto-1-propanol; they revealed a significant cytotoxicity in cervical, prostate, and colon cancer cells without affecting the normal cells [34].

Taking into a consideration the current scenario of resistance to chemotherapy and its adverse impacts on patients with leukemia, it is very important to conduct studies that evaluate the potential of nanoparticulated active compounds. In this sense, the present study evaluated the in vitro cytotoxic and immunomodulatory activity of bismuth NPs in myeloid leukemia cells.

In the present study, we obtained stable NPs of hydrated bismuth oxide and demonstrated their high and promising leishmanicidal and antileukemic activities.

## 2. Materials and Methods

All reagents were of the ACS grade or higher and were purchased from Merck (Darmstadt, Germany) unless otherwise specified.

### 2.1. Preparation of Hydrated Bismuth Oxide Nanoparticles (BiNP)

Ultra-high-purity freshly deionized water (ρ ≥ 18.0 MΩcm) was used throughout the experiments. The synthesis was realized according to a procedure developed by us [35,36] by a careful hydrolysis of BiCl_3_ solution in a great molar excess (>250×) of water containing monosodium salts of 2-mercapto-1-ethanesulfonic and 3-mercapto-1-propanesulfonic acids (MES or MPS, respectively; TCI Europe, Zwijndrecht, Belgium) as a stabilizing ligand. The obtained transparent slightly opalescent sols were purified by dialysis until they were essentially chloride-free. The dialysis was performed in water, using cellulose membrane tubing (Sigma-Aldrich, St. Louis, MO, USA, product no D9777) according to standard procedures.

### 2.2. Bismuth Assay

The quantification of Bi was performed by atomic emission spectroscopy using an Agilent microwave plasma spectrophotometer (4100 MP-AES, Agilent Technologies, Santa Clara, CA, USA) equipped with an SPS 3 autosampler (Agilent), a standard torch, an Inert OneNeb nebulizer, and a double-pass glass cyclonic spray chamber (Agilent). Nitrogen was obtained from the air using a nitrogen generator (Agilent 4107). Time intervals of 15 s for uptake time and 20 s for the torch stabilization were set prior to measuring the samples. The analysis was performed at detection wavelengths of 222.825, 223.061, 289.798, and 306.772 nm; the read time was set to 5 s. The spectral intensity was obtained as a mean of 5 replicate readings per sample. 

Solutions for the calibration curve (100, 80, 60, 40, 30, 20, and 10 mg/L) were obtained by the dilution of TraceCERT^®^, a 1000 mg/L Bi standard with 1.0 M HCl. The sample solutions were prepared by the dissolution of 100 µL of BiNP hydrosol in 500 µL of concentrated HCl and brought to volume of 10 mL with 1.0 M HCl.

### 2.3. Nanoparticle Characterization

#### 2.3.1. Dynamic Light Scattering (DLS)

Particle size and ζ-potential measurements were performed with a Zetasizer Nano (Malvern Instruments Inc., Westborough, MA, USA) using Bi_2_O_3_·nH_2_O hydrosols in appropriate polystyrene cuvettes.

#### 2.3.2. Transmission Electron Microscopy (TEM)

TEM micrographs were obtained with an FEI Tecnai G12 Spirit-Twin transmission electron microscope with an LaB_6_ source and operating with an acceleration voltage of 120 kV, equipped with a bottom-mounted FEI Eagle-4k CCD camera (Eindhoven, The Netherlands).

### 2.4. Cell Culture

The leishmania parasite strains *Leishmania* (*Leishmania*) *amazonensis* (IFLA/BR/67/PH8) and *Leishmania* (*Viannia*) *guyanensis* (MHOM/BR/75/M4147) were obtained from the Roswell Park Memorial Institute and grown at 25 °C in the RPMI 1640 medium (Gibco, Rockville, MD, USA), supplemented with inactivated 10% fetal bovine serum (FBSi; LGC Biotechnology, São Paulo, Brazil) and 50 µg/mL gentamicin (Novafarma, Brazil).

Murine peritoneal macrophages were grown in 96-well plates (10^5^ cells/mL) in RPMI medium, supplemented with FBSi, and incubated in a 5% CO_2_ incubator (Series II Water Jacket CO_2_ Incubator, Thermo Scientific, Waltham, MA, USA) at 37 °C.

The human leukemia cell lines HL60 (ATCC^®^ CCL-240™—Acute promyelocytic leukemia) and K562 (ATCC^®^ CCL-243TM—Chronic myeloid leukemia) were cultured in RPMI medium, while Vero cells (renal epithelial cell line derived from the African green monkey, *Chlorocebus* sp.) were cultured in DMEM, Dulbecco’s modified Eagle’s medium (Gibco). All media were supplemented with FBSi (Gibco), 100 μg/mL of penicillin, and 100 μg/mL of streptomycin. 

All experiments were performed in triplicate.

#### 2.4.1. Biological Assays in Promastigote Forms

Promastigotes were grown in 96-well plates (2 × 10^6^ promastigotes/mL) in RPMI supplemented with 10% FBSi and treated for 24, 48, and 72 h with different concentrations of BiNPs (870.0 to 50.0 nM), Glucantime^®^ (Sanofi Medley Farmacêutica Ltd.a., Suzano (SP), Brazil; 33.0 to 2.06 mM), and MES (4.35 to 0.2 mM) obtained by serial dilutions of the corresponding stock solutions. The wells with untreated promastigotes were maintained as controls. Biological activity was determined by quantifying viable promastigotes in a hemocytometer, using a Trypan Blue dye and an optical microscope (Nikon Eclipse E200, Tokyo, Japan) with 400× magnification. The data obtained were expressed as half maximum inhibitory concentrations (IC_50_) [21].

All experiments were performed in triplicate. The value of IC_50_ were determined using linear regression in a GraphPad Prism software (v.8.0).

#### 2.4.2. Biological Assays in Amastigote Forms

The evaluation of the leishmanicidal activity of BiNPs was performed in murine peritoneal macrophages infected with *Leishmania* spp. promastigotes in a 1:10 ratio (10^5^ cells:10^6^ parasites) on glass coverslips in 24-well plates with RPMI culture medium supplemented with 10% FBSi and incubated in a 5% CO_2_ incubator at 37 °C for up to 2 h. Subsequently, the infected cells were treated for 24, 48, and 72 h with different concentrations of BiNPs (870.0 to 100.0 nM), Glucantime^®^ (33.0 to 2.06 mM), and MES (4.35 to 0.4 mM) obtained by serial dilutions of the corresponding stock solutions. The untreated samples were used as a negative control. Then, the coverslips were fixed and stained every 24 h by the Quick Panoptic method (Laborclin^®^, Paraná, Brazil) and analyzed by optical microscopy. The percentage of infected cells was determined by randomly counting 100 infected and uninfected cells in each coverslip [21].

All experiments were performed in triplicate. The value of IC_50_ were determined using linear regression in GraphPad Prism software (v.8.0).

#### 2.4.3. Cytotoxicity Assays

##### Resazurin Assay

Murine peritoneal macrophages were treated with different concentrations of BiNPs (870.0 to 50.0 nM), Glucantime^®^ (33.0 to 2.06 mM), and MES (4.35 to 0.2 mM). Wells without cells were kept blank and wells with cells without treatment were kept as controls. The cell viability at the intervals of 24, 48, and 72 h was evaluated by colorimetry using a resazurin sodium salt [37]. In total, 10 µL of the resazurin stock solution in phosphate buffer saline (PBS) (4 mg/mL) was added into each well and the plates were incubated again for 12 h at 37 °C. The absorbance was read on an Elx800^™^ spectrophotometer (BIO-TEK^®^, Winooski, VT, USA) at a wavelength of 590 nm. The data were normalized according to the following formula [38]: $$Cell\;viability=\frac{\left|Sample\right|-\left|Blank\right|}{\left|Control\right|-\left|Blank\right|}\times 100$$

##### MTT Assay

The 3-(4,5-dimethylthiazol-2-yl)-2,5-diphenyltetrazolium bromide (MTT) assay was employed for viability assessment. The cytotoxicity of Bi-MES and MES was assessed using K562, HL60, and non-cancerous Vero cell lines. The cells were cultured in 96-well culture plates (104 cells/well) containing 0.2 mL per well of RPMI and DMEM medium, respectively; supplemented with 10% FBSi, penicillin–streptomycin, and fungizone (100 μg/mL). The cells were then treated with Bi-MES and MES at concentrations ranging from 0.18 µg/mL to 3 µg/mL. Sterile PBS containing 0.5% DMSO served as the negative control, while 100% DMSO was used as the positive one. The plates were incubated for 24 h at 37 °C in a 5% CO_2_ atmosphere. After the incubation period, the supernatant was removed, and 10 μL of a 5 mg/mL MTT solution and 100 μL of phenol red-free RPMI medium were added to each well, followed by incubation at 37 °C for 4 h. The reaction was terminated by the addition of 100 μL of DMSO.

The cell growth was assessed by spectrophotometry measuring the absorbance at 570 nm on a microplate reader. The relative viability of the cells was estimated using the following equation:$$Cell\;viability=\frac{A570\;of\;the\;treated\;sample}{A570\;of\;the\;untreated\;sample}\times 100$$

#### 2.4.4. Colony Formation Assay

Clonal expansion of cancer cells after treatment with BiNPs was evaluated using a colony formation assay. Approximately 500 HL60 cells were cultured in 6-well plates with BiNPs at the concentration of 0.37 μL/mL in a semi-solid cell suspension medium (MethoCult 4230, StemCell Technologies Inc., Vancouver, BC, Canada). The colonies were detected after 8–10 days of culture by addition of the MTT reagent (1 mg/mL), and the colony count (>50 cells per colony) was performed using an Image J 1.54g quantification software (NIH, Bethesda, MD, USA), as previously described [39]. Briefly, the images were captured using an inverted microscope at 40× magnification and imported into ImageJ for processing. Brightness and contrast were adjusted, and the scale was set using known dimensions. The images were converted to binary to differentiate colonies. The ‘Watershed’ algorithm was applied to separate touching colonies, and the ‘Analyze Particles’ feature was used to quantify the colonies, providing the count and size data. The results obtained were then compiled for statistical analysis.

### 2.5. Data Analysis

The data obtained were analyzed using GraphPad Prism software (v.8.0). All quantitative variables were expressed as the mean ± standard deviation or median. The Chi-square test or Fisher’s exact test for categorical variables were also employed. ANOVA tests were utilized to assess the relative cytotoxicity of the BiNPs. Values with *p* < 0.05 were considered statistically significant.

## 3. Results

### 3.1. Obtaining and Characterization of Nanoparticles

Hydrosols of hydrated bismuth oxide were obtained as a stable yellowish slightly opalescent liquid by a procedure proposed by us. In the case of the MPS ligand, the resulting sol had a darker color. The bismuth content in both sols was determined by MP-AES; the nanoparticle size and ζ-potential measurements were performed by the DLS technique, and the size was further confirmed by TEM. The experimental results are presented in Table 1. TEM microimages of the materials are demonstrated in Figure 1. 

It appeared that the hydrosol containing the MPS ligand was not sufficiently stable, since it started to precipitate after approximately one week; therefore, only MES stabilized samples were selected for further evaluation of their biological activity.

### 3.2. Leishmanicidal Activity of Bismuth Nanoparticles

During the entire incubation period, BiNPs demonstrated excellent leishmanicidal activity in promastigote forms; IC_50_ < 46.0 nM was found for both *L.* (*L.*) *amazonensis* and *L.* (*V.*) *guyanensis*. Such values of the half maximum inhibitory concentrations were several orders of magnitude lower than those determined for the currently used first-line pentavalent antimonial drug Glucantime^®^ (IC_50_ > 45.10 mM for *L.* (*L.*) *amazonensis* and IC_50_ > 27.64 mM for *L.* (*V.*) *guyanensis*). The MES ligand showed significantly lower toxicity to both parasite species (IC_50_ > 150.0 µM and IC_50_ > 9.0 µM for *L.* (*L.*) *amazonensis* and *L.* (*V.*) *guyanensis*, respectively).

It is interesting to note that BiNPs demonstrated a better efficacy of parasite inhibition for *L.* (*L.*) *amazonensis* at the concentration of 870.0 nM (*p* < 0.005). More diluted samples started to lose their efficiency. Even the maximum applied concentration of Glucantime^®^ (33.0 mM) was less efficient in parasite inhibition, despite the active substance concentration value being approximately 38,000 times higher (*p* <0.05) (Figure 2).

In the case of *L.* (*V.*) *guyanensis*, significant reduction in the parasite viability was observed for BiNPs concentrations of 870.0–210.0 nM (*p* < 0.005). The comparison with Glucantime^®^ demonstrated that the most concentrated BiNP sample (870.0 nM) induced a greater reduction in the parasite viability only in the first 24 h (*p* < 0.05) (Figure 2).

A lower percentage of cells infected with *L.* (*L.*) *amazonensis* amastigote forms (<50%) was observed in murine peritoneal macrophages (Figure 3) exposed to BiNPs when compared to the negative control (>60%) (*p* < 0.001 in both cases). A comparison with the Glucantime^®^ (33,000 µM) and MES ligand (4350 µM) demonstrated that the same decrease in the infectivity rate (*p* < 0.001) can be achieved with only a 0.87 µM concentration of BiNPs in 48 and 72 h of treatment.

For *L.* (*V.*) *guyanensis*, the BiNPs revealed the highest suppression of the percentage of infected cells (52%) within 72 h when compared to the negative control (70%) (*p* < 0.05 in both cases). However, there was no significant difference (Figure 3) in the percentage of infected macrophages observed for the treatment with BiNPs and much more concentrated Glucantime^®^ and MES solutions.

Our results also demonstrated that BiNPs, Glucantime^®^, and MES were practically non-toxic to murine peritoneal macrophages at the concentration applied, and the cell viability values were above 90%, 90%, and 73%, respectively.

### 3.3. Cytotoxic Activity of Bismuth Nanoparticles

Bismuth nanoparticles (BiNP) exhibited significant cytotoxic activity in the HL60 (*p* < 0.0001) and K562 cell lines (*p* < 0.001) at concentrations ranging from 0.18 µg/mL to 1.5 µg/mL (0.86–7.17 µM). The activity compared to the negative control (untreated cells) is shown in Figure 4A,C, respectively. On the contrary, the MES ligand (control) demonstrated significant cytotoxic effects (*p* < 0.01) at concentrations ranging from 0.37 to 1.5 µg/mL (2.85–9.14 µM) in HL60 cells (Figure 4B), although that activity was much lower when compared to the BiNPs. No cytotoxic activity of the control was observed in K562 cells (Figure 4D) at all tested concentrations.

BiNPs revealed a certain cytotoxicity for Vero cells at all tested concentrations (*p* < 0.05) (Figure 4E), while the MES ligand control was found to be non-cytotoxic for those cells (Figure 4F). However, the cytotoxicity of BiNPs in normal Vero cells was lower when compared to that in the cancerous HL60 and K562 cells. On average, the BiNPs reduced the viability of HL60 cells by 49% and K562 cells by 38%, whereas the average reduction of only 27% was found in for Vero cells.

### 3.4. Colony Formation Analysis

The effect of bismuth nanoparticles and the MES ligand on HL60 leukemic cell colony formation is shown in Figure 5. The results demonstrated a significant reduction (79% and 62%, respectively; Figure 5A,B) in HL60 cell colony formation when compared to the untreated cells (*p* < 0.0001). Additionally, a size comparison of the cell colony treated with the MES ligand and the untreated one (control) indicates the ligand effectiveness with respect to the control. The observation is evident from the colony images shown in Figure 5C,D.

## 4. Discussion

Currently, the treatments available for cutaneous leishmaniasis have limited effectiveness leading to the necessity of looking for new active principles. Thus, taking into account the biological activity of bismuth reported in the literature and the advantages nanotechnology, this paper presents data on the synthesis and characterization of bismuth nanoparticles (BiNPs) together with data on their cytotoxicity, antileukemic, and leishmanicidal activity.

Colloidal bismuth hydroxide was first obtained in 1902 [40] by the slow hydrolysis of a 0.23% bismuth nitrate solution followed by dialysis. To stabilize the colloid, a treatment with sodium protalbinate was later proposed [41]. However, the impossibility to remove the protein from the sol strongly limited interest in the material. Another way to obtain colloidal Bi(OH)_3_ involved a peptization of freshly precipitated hydroxide with a 70 °C hot strong alkaline sucrose solution [42]. Again, the sol was only stable at extremely high pH values.

We succeed in obtaining stable hydrosols of Sb and Bi by a careful hydrolysis of the corresponding metal chlorides in a high molar excess of water in the presence of stabilizing ligands [35,36].
2 BiCl_3_ + (3 + *n*) H_2_O = Bi_2_O_3_ × *n*H_2_O ↓ + 6 H^+^ + 6 Cl^−^

In the case of bismuth, the ligands of choice were short-chain mercaptoalkylsulfonates chosen because of the greater thermodynamic stability of Bi–S bonds with respect to Bi–O bonds; additionally, the Bi–S bonds are known to be more labile, providing more availability of the metal in biological applications [6].

The mean particle size of BiNPs and their ζ-potential were measured by DLS (see Table 1) using the fresh hydrosols in appropriate polystyrene cuvettes. The values measured were in good agreement with the TEM data, which also demonstrated the regular shape of the nanoparticles (see Figure 1). The magnitude of the ζ-potential in the range of −35–−38 mV (Table 1) indicates that the surface of hydrated bismuth oxide nanoparticles is protected by sulfonate anions (negative sign) and gives evidence of a high stability of the sols due to the effective electrostatic repulsion of the adjacent particles. However, the sols obtained revealed different stability with time. While the MES-containing BiNPs were stable for several months, the MPS treatment was not so successful and the bismuth hydroxide precipitated after several days. The reasons behind that difference are still to be found in the forthcoming investigations.

Therefore, the hydrosol of bismuth hydroxide stabilized with MES was used in further investigations of biological activity. 

Bismuth is a metal considered relatively non-toxic to humans, although it is capable of providing a powerful antimicrobial activity [12,19]. Recently, several studies have been carried out to evaluate the leishmanicidal activity of Bi(III) and Bi(V) complexes. The former demonstrated promising results in *L. major* promastigotes with the values of IC_50_ < 0.18 µM being harmless to human fibroblasts [43], and the latter showed a high parasite mortality (>97%) and IC_50_ = 2.78 µg/mL values against promastigote forms of *L. tropica* without being cytotoxic for human lymphocytes [44].

As expected, the BiNPs obtained by us were not cytotoxic to murine peritoneal macrophages (>90% cell viability), nor was Glucantime^®^. Those observations were in good agreement with the data reported in the literature [43,44,45]. The former demonstrated the absence of toxicity to human fibroblasts (100% cell viability) for five bismuth (III) coordination compounds and the latter suggested that Bi(V) compounds were significantly non-toxic (approximately 60% viability) at concentrations of about 25µM. 

Regarding the leishmanicidal activity, it was observed that BiNPs showed better efficacy in both the promastigote and amastigote forms of *L.* (*L.*) *amazonensis*, with an IC_50_ < 46.0 nM value for the promastigotes and a percentage of infected macrophages < 50%. Therefore, the results obtained for the nanoparticle treatment of the promastigote forms of *L.* (*L.*) *amazonensis* are several orders of magnitude better than previous data reported for coordination compounds of bismuth that showed IC_50_ values of 8.5 µM and 1.07 µM for Bi(V) [46] and Bi(III) [47] complexes, respectively. To date, there have been no publications dedicated to investigation of the leishmanicidal activity of bismuth-based compounds or materials towards *L.* (*V.*) *guyanensis*; and therefore, the present paper is the first report in the area. Thus, the results obtained for Bi-containing nanomaterials in this study demonstrate that bismuth-based materials could be a promising alternative to currently used pentavalent antimonials, owing to their greater leishmanicidal efficacy and lower toxicity in relation to Sb(V) derivatives [19].

The Bi-containing nanoparticles have also been studied as a promising option for leukemia treatment, offering advantages such as low toxicity and overcoming chemotherapy resistance [32,33]. Among many different heavy metals, bismuth is considered less toxic to the human body [32,33]. Furthermore, bismuth compounds usually require very low levels of pH to be stable in an aqueous phase. Hence, bismuth can become cytotoxic to human cells at higher pH values, like those found under normal physiological conditions [48]. However, the strategy of using bismuth nanoparticles allows this metal to be applied in aqueous media (hydrosol) even at higher pH values; thus, optimizing the concentrations required to achieve a biological effect. In other words, bismuth in the nanoparticle form can be delivered to the target cells at physiological pH and will exhibit cytotoxicity to cancer cells, while causing less harm to the healthy ones [32,33]. Previously, lipophilic bismuth nanoparticles were reported to exhibit significant cytotoxicity to MCF-7 breast cancer cells and relatively low toxicity to non-cancerous MCF-10A ones [49]. Later, it was found that bismuth (III) oxide nanoparticles (Bi_2_O_3_ NPs) showed significant cytotoxicity towards lung (A549) and liver cancer (HepG2) cells. At the same time, no cytotoxic activity was observed in non-cancerous primary hepatocyte cells, demonstrating the selectivity of bismuth (III) oxide nanoparticles [50].

The present study revealed the anticancer potential of bismuth nanoparticles by showing selective cytotoxic activity towards myeloid leukemia cell lines, while affecting non-cancerous cells to a lesser extent. Additionally, the BiNPs were able to rapidly inhibit the clonal proliferation of both HL60 and K562 cells. Our finding is in good agreement with recent studies indicating the potential of bismuth nanoparticles for different biomedical applications, including cancer therapy [4,51,52]. It is worth noting that BiNPs could intrinsically degrade and dissolve at lower pH values. Oxidative and acidic microenvironment in cancer cells with a typical pH ranging from 6.4 to 7.2 may induce a biodegradation of the nanoparticles [48], which could become relevant from a therapeutic viewpoint, since both bismuth ions and their agglomerates may exert their antiproliferative effects [48].

The BiNPs demonstrated significant cytotoxic activity against K562 and HL60 cells at all evaluated concentrations. On the other hand, the MES ligand control also exhibited a certain cytotoxic activity remarkable at the higher concentration of 3 µg/mL (18.27 μM) for the K562 and starting from 0.37 µg/mL (2.25 μM) for the HL60 cell line. Although some cytotoxicity of the BiNPs was also observed in Vero cells, this was lower when compared to the cancerous HL60 and K562 cell lines. Treatment with the bismuth nanoparticles also had a significant effect on inhibiting and reducing colony formation in HL60 cells. These results indicate that bismuth nanoparticles have the potential for an inhibitory effect on the clonal expansion of cancer cells.

## 5. Conclusions

The results of the present study demonstrate that hydrated bismuth oxide nanoparticles exhibit a very high efficiency (several orders of magnitude higher than currently used antimonial drug Glucantime^®^) and significantly reduce the percentage of infected cells in *Leishmania* parasite infection, being non-cytotoxic to murine peritoneal macrophages at the same time. Similarly, the BiNPs appear to be a promising option for leukemia treatment and practically non-toxic to normal Vero cells. The nanoparticulated form of the active compound has clear advantages over molecular ones, since it can produce a high local concentration of the API within the target cell upon a single endocytosis act. This property could be regarded as a precondition to overcome the resistance of parasites and tumor cells to molecular forms of the drugs. It was not possible to perform apoptosis tests to analyze the mechanism of action of the bismuth nanoparticles in our study. However, some published data [8,53] suggest that the cytotoxic activity of bismuth could primarily be related to the inhibition of lipases and glycosidases and the production of reactive oxygen species that damage mitochondrial membranes. It is worth noting that the stability of the BiNP hydrosol opens the possibility of a patient-friendly topical drug administration instead of a systemic one, which is of particular importance for the treatment of cutaneous leishmaniasis. In a nutshell, the results indicate that BiNPs are highly efficient antileishmanial active compounds that also exhibit a remarkable antiproliferative potential without causing significant cytotoxicity in non-cancerous cells at the tested concentrations. However, further research is needed to better understand the mechanisms of the action and safety profile of bismuth-based nanoparticles as well as to ascertain leishmanicidal and antileukemic activity in vivo.

## Figures and Tables

**Figure 1 pharmaceutics-16-00874-f001:**
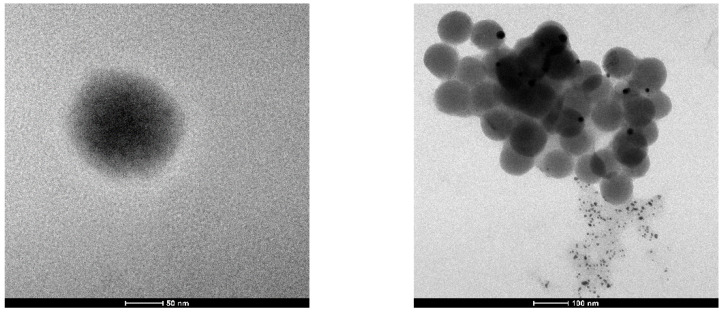
TEM micrographs of the BiNP samples: stabilized with MES (**left**) and MPS (**right**).

**Figure 2 pharmaceutics-16-00874-f002:**
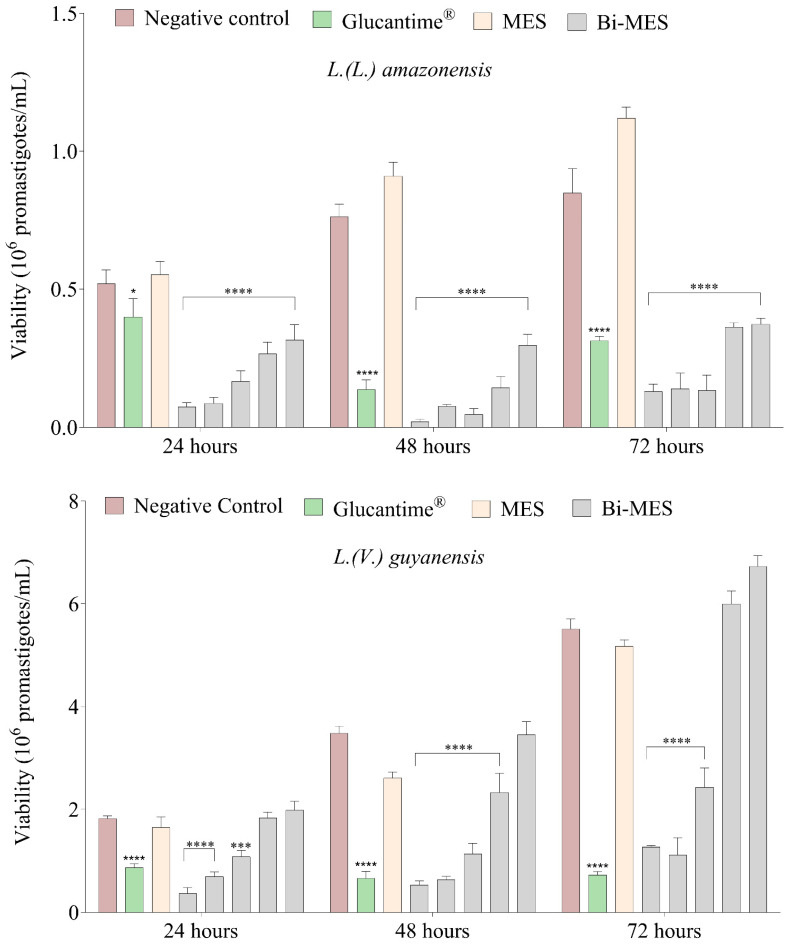
Viability of *L.* (*L.*) *amazonensis* and *L.* (*V.*) *guyanensis* parasites. Untreated promastigotes (negative control), treated with Glucantime^®^ 33,000 μM, MES 4350 μM, and different concentrations of BiNPs (from left to right for each series): 0.87 μM, 0.43 μM, 0.21 μM, 0.10 μM, and 0.05 μM. * *p* < 0.05; *** *p* < 0.001; **** *p* < 0.0001 (ANOVA).

**Figure 3 pharmaceutics-16-00874-f003:**
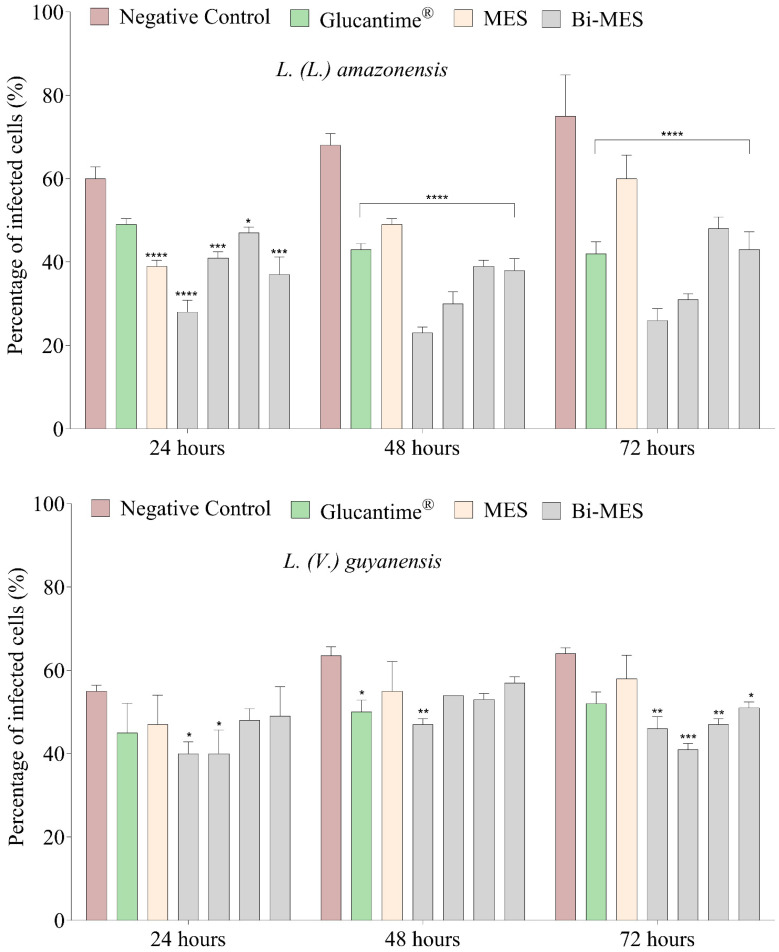
Percentage of infected murine peritoneal macrophages infected by *L.* (*L.*) *amazonensis* and *L.* (*V.*) *guyanensis*. Untreated macrophages (negative control), treated with Glucantime^®^ 33,000 μM, MES 4350 μM, and different concentrations of BiNPs (from left to right for each series): 0.87 μM, 0.43 μM, 0.21 μM, 0.10 μM, and 0.05 μM. * *p* < 0.05; ** *p* < 0.01; *** *p* < 0.001; **** *p* < 0.0001 (ANOVA).

**Figure 4 pharmaceutics-16-00874-f004:**
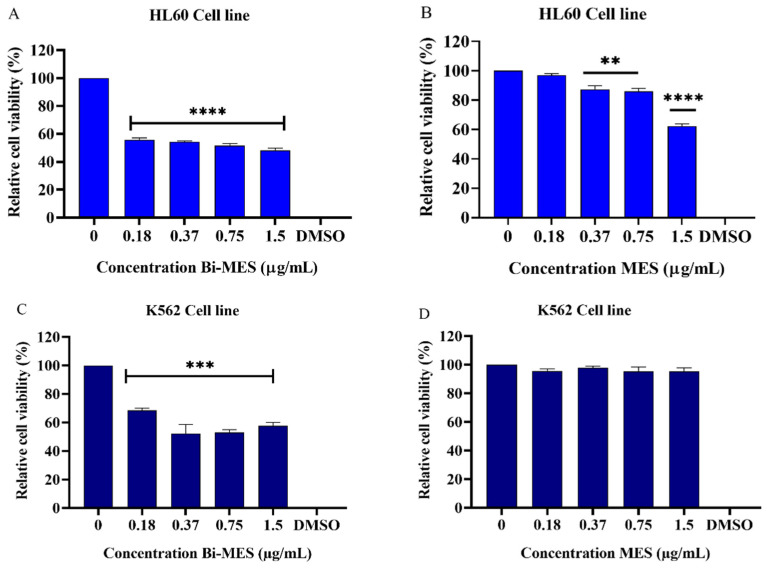
Cytotoxic activity of bismuth nanoparticles (Bi-MES) and the MES ligand. The cells lines HL60 and K562 were treated with BiNPs for 24 h at concentrations of 0.18–1.5 µg/mL (0.86–7.17 µM) (**A**,**C**) and the MES ligand control at concentrations of 0.37 to 1.5 µg/mL (2.85–9.14 µM) (**B**,**D**), respectively. The cytotoxicity of BiNPs (**E**) and MES (**F**) to Vero cells at concentrations of 0.37–3.0 µg/mL (1.77–14.36 µM for BiNP and 2.25–18.27 µM for MES) for 24 h. *: indicates significant difference compared to the control (untreated cells), where * *p* < 0.05; ** *p* < 0.01; *** *p* < 0.001; **** *p* < 0.0001 (ANOVA).

**Figure 5 pharmaceutics-16-00874-f005:**
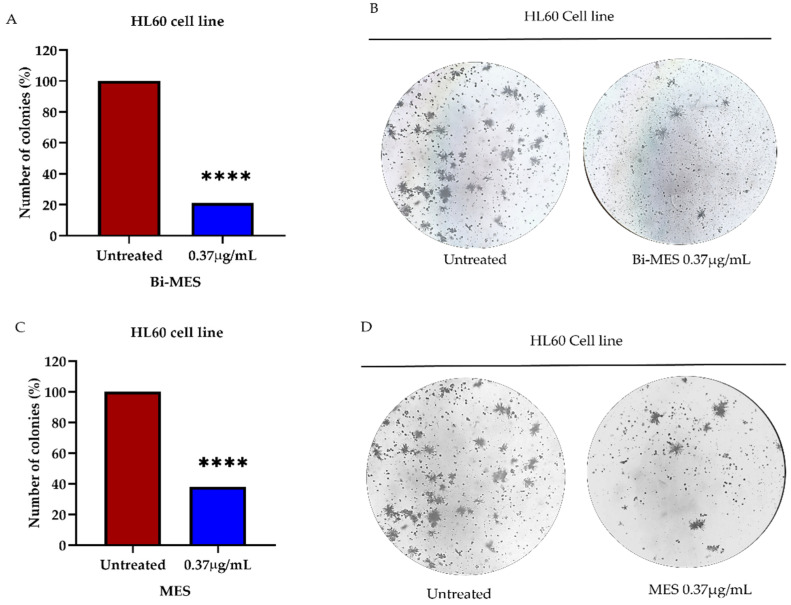
Effect of the BiNPs and MES ligand on HL60 cell colony formation. Quantification of colonies after the treatment with 0.37 μg/mL (1.77 µM) of BiNPs (**A**) and 0.37 μg/mL (2.25 µM) MES (**C**) as well as representative optical microscopy images (40×) of the same (**B**,**D**). Statistical significance is denoted by asterisks: **** *p* < 0.0001.

**Table 1 pharmaceutics-16-00874-t001:** Concentration, particle size, and ζ-potential of Bi_2_O_3_·nH_2_O nanoparticles.

Stabilizing Ligand	[Bi^3+^] Concentration, mM	Particle Size, nm	ζ-Potential, mV
MES	25.93 ± 0.22	70 ± 23	−38.1 ± 5.53
MPS	20.31 ± 0.41	54 ± 11	−35.1 ± 4.28

## Data Availability

Data are available upon request.

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
