# Peer review of "A Second Wind for Inorganic APIs: Leishmanicidal and Antileukemic Activity of Hydrated Bismuth Oxide Nanoparticles"

_pharmaceutics, 2024, doi:10.3390/pharmaceutics16070874_

Round 1
Reviewer 1 Report
Comments and Suggestions for Authors
I find this an interesting and relevant paper, however there are a number of key issues that detract from the research presented.
Firstly, I feel that the two areas included in this manuscript should be separated into two publications. From my understanding of the area the work on leishmania is truly novel I only know of this paper that has limited similarity to the work presented Islam A, et al 2024. Susceptibility of Leishmania to novel pentavalent organometallics: Investigating impact on DNA and membrane integrity in antimony(III)-sensitive and -resistant strains. Drug Dev Res. doi: 10.1002/ddr.22194.
Leukemias are not my area so can't comment on the novelty of that part of the manuscript.
Secondly the presentation of the leishmania data is poor- this needs to be revisited and modified. Fig 2 would be better as a series of panels with concentration vs viability - and show these curves at three time points this makes the data easier to interpret - also the graph for L guyanensis seems to be missing appropriate labels and check spelling of the species. I also find it difficult to extract the data from figure 3. In addition to that the numbers of replicates are not given, and the methods used to determine the IC50 is not explained in the data analysis section. Figure 1 is also confusing, using the scale bars the size of particles seems to be different from that presented in the table different the single particle in the left had side panel has a size of approx. 120-150nm.
The term infection rate is also used - I'm not sure that is what you are presenting - your data is % infection?
Another methodological query- what was the method used to determine viable promastigotes - line 179 - this was performed by microscopy but were any stains used?
The data for the cell line cytotoxicity was much better presented and would be a good format to use for the leishmania, but still needs complete legends re replicates.
Other issues to address:
1) Would be good to include a little more info re the fact that these materials have V low levels of toxicity
2) The manuscript needs further proofreading for example line 34 'the colorimetry'? line 67 'provoke' should be cause? line 84 'Up to date'
Overall, I feel that this paper should focus on one target 'leishmania' with improved presentation of data.
Comments on the Quality of English Language
find this an interesting and relevant paper, however there are a number of key issues that detract from the research presented.
Firstly, I feel that the two areas included in this manuscript should be separated into two publications. From my understanding of the area the work on leishmania is truly novel I only know of this paper that has limited similarity to the work presented Islam A, et al 2024. Susceptibility of Leishmania to novel pentavalent organometallics: Investigating impact on DNA and membrane integrity in antimony(III)-sensitive and -resistant strains. Drug Dev Res. doi: 10.1002/ddr.22194.
Leukemias are not my area so can't comment on the novelty of that part of the manuscript.
Secondly the presentation of the leishmania data is poor- this needs to be revisited and modified. Fig 2 would be better as a series of panels with concentration vs viability - and show these curves at three time points this makes the data easier to interpret - also the graph for L guyanensis seems to be missing appropriate labels and check spelling of the species. I also find it difficult to extract the data from figure 3. In addition to that the numbers of replicates are not given, and the methods used to determine the IC50 is not explained in the data analysis section. Figure 1 is also confusing, using the scale bars the size of particles seems to be different from that presented in the table different the single particle in the left had side panel has a size of approx. 120-150nm.
The term infection rate is also used - I'm not sure that is what you are presenting - your data is % infection?
Another methodological query- what was the method used to determine viable promastigotes - line 179 - this was performed by microscopy but were any stains used?
The data for the cell line cytotoxicity was much better presented and would be a good format to use for the leishmania, but still needs complete legends re replicates.
Other issues to address:
1) Would be good to include a little more info re the fact that these materials have V low levels of toxicity
2) The manuscript needs further proofreading for example line 34 'the colorimetry'? line 67 'provoke' should be cause? line 84 'Up to date'
Overall, I feel that this paper should focus on one target 'leishmania' with improved presentation of data.
Author Response
The Authors would like to thank the Reviewer for his/her time dedicated to evaluation of our manuscript.
I find this an interesting and relevant paper, however there are a number of key issues that detract from the research presented.
Firstly, I feel that the two areas included in this manuscript should be separated into two publications. From my understanding of the area the work on leishmania is truly novel I only know of this paper that has limited similarity to the work presented Islam A, et al 2024. Susceptibility of Leishmania to novel pentavalent organometallics: Investigating impact on DNA and membrane integrity in antimony(III)-sensitive and -resistant strains. Drug Dev Res. doi: 10.1002/ddr.22194.
Leukemias are not my area so can't comment on the novelty of that part of the manuscript.
Response: The manuscript properly covers the research conducted by Prof. Frézard – Dr Demicheli’s group who study biological activity of Bi complexes making reference to papers that are relevant to the topic under discussion. The article mentioned is one of the newest publications of the group; however, it is devoted to research that is not so relevant to our investigation. Therefore, that article was not cited.
Secondly the presentation of the leishmania data is poor- this needs to be revisited and modified. Fig 2 would be better as a series of panels with concentration vs viability - and show these curves at three time points this makes the data easier to interpret - also the graph for L guyanensis seems to be missing appropriate labels and check spelling of the species. I also find it difficult to extract the data from figure 3.
Response: The figures were revised.
In addition to that the numbers of replicates are not given, and the methods used to determine the IC50 is not explained in the data analysis section.
Response: The information was provided.
Figure 1 is also confusing, using the scale bars the size of particles seems to be different from that presented in the table different the single particle in the left had side panel has a size of approx. 120-150nm.
Response: The particle size presented in the Table were obtained by DLS, while the Figure 1 shows a TEM micrograph. The data were in a very good agreement for the Bi-MES nanoparticles (used in further experiments). The MPS stabilised nanoparticles might probably grow a bit more with time, since the DLS was performed on freshly obtained sols and TEM studies were performed later.
The term infection rate is also used - I'm not sure that is what you are presenting - your data is % infection?
Response: Corrected
Another methodological query- what was the method used to determine viable promastigotes - line 179 - this was performed by microscopy but were any stains used?
Response: The determination was performed using a Trypan Blue dye https://www.sigmaaldrich.com/BR/pt/product/sigma/t8154
The data for the cell line cytotoxicity was much better presented and would be a good format to use for the leishmania, but still needs complete legends re replicates.
Other issues to address:
1) Would be good to include a little more info re the fact that these materials have V low levels of toxicity
Response: The observed lower cytotoxicity of bismuth nanoparticles in non-cancerous cells suggests their selective action in targeting only cancerous cells. The underlying mechanisms remain unclear, but this selectivity warrants further investigation for its potential as a targeted cancer therapeutic platform.
2) The manuscript needs further proofreading
for example line 34 'the colorimetry'?
Response: The spelling of the term “colorimetry” is absolutely correct. We invite the reviewer to consult either Merriam-Webster dictionary (M-WD) for the American spelling or Oxford English Dictionary (OED) for the British one.
line 67 'provoke' should be cause?
Response: The use of the word “provoke” is absolutely correct. “Provoke” means “to give rise to” (OED), “to provide the needed stimulus for” (M-WD). The verb cause is a synonym. It is the Authors’ preference which word to use according to own style and the context.
line 84 'Up to date'
Response: The adverb “up to date” is spelled correctly (OED).
Overall, I feel that this paper should focus on one target 'leishmania' with improved presentation of data.
Response: The Authors designed the experiments and decided to present the results in a way described in the manuscript. There is no need to split the manuscript unnecessarily in several pieces.
Reviewer 2 Report
Comments and Suggestions for Authors
The manuscript addresses an important issue, but needs improvement 1. The introduction is written quite nicely. I didn’t really understand the reasoning about the connection between metal density and stability. If we talk about density, then the density of lead is 10% greater. In general, the authors provide a link to the authoritative work of Dr. De Marcillac, dedicated to the discovery of the radioactive isotope of bismuth. As for the use of bismuth compounds for the treatment of oncological diseases, the authors are disingenuous and deliberately mislead readers. In fact, there are experimental studies showing the effectiveness of bismuth compounds against certain forms of cancer cells. There is no real bismuth drug approved by the Ministry of Health of any country for the treatment of cancer and there probably won’t be for a long time. Probably, the authors need to slightly adjust the categoricalness of their statements. 2. Materials and methods. The authors write: “Ultra-high-purity freshly deionized water (ρ ≥ 18.0 MΩcm)…”. Judging by the resistivity, it's just reverse osmosis water. Please correct the categorical statements. Section 2.4.3 Cytotoxicity assays - there is probably no point in describing generally accepted methods in detail; you can simply refer to the original source. All researchers are aware of how to use these methods. 3. Dynamic light scattering. DLS, without major modifications, in principle cannot determine hundredths of nanometers, as for example stated in Table 1 (69.91 nm or 53.76 nm). The authors need to describe what approach allowed them to achieve such amazing accuracy. What the authors changed in the optical design of the device and the mathematics that calculates the autocorrelation function. 4. The marks in Figure 1 are not readable. Please fix it! 5. The labels of the axes of Figure 2 are not readable. Please fix it! Without understanding what is shown in the figure, it is very difficult to review a manuscript. 6. Figure 3 presents data on the rate of infection of mouse peritoneal macrophages infected with L. (L.) amazonensis and L. (V.) guyanensis. Looking at the data, we can say that there are no effects! To argue otherwise, the authors should conduct a statistical study and present the data as means plus or minus the standard error of the mean. Place the appropriate stars. 7. Comparison of leukemic cells and monkey epithelial cells is not correct in essence. Authors should provide an explanation in the manuscript why this particular model was chosen. 8. The discussion is quite massive and is written, by and large, clearly, although not always appropriately. I do not understand why the authors do not build on their success in the discussion and do not try to extend their discovery to other areas. It can be assumed that the nanoparticle preparations developed by the authors will be in demand for protecting the surfaces of food production, developing antibacterial coatings, packaging, and other things. As an example, I advise you to read Chapter 8 of Article 10.3367/UFNe.2023.09.039577.
Author Response
First of all, the Authors would like to draw the Reviewer’s attention to inappropriate style of his/her comments.
The manuscript addresses an important issue, but needs improvement
1. The introduction is written quite nicely. I didn’t really understand the reasoning about the connection between metal density and stability. If we talk about density, then the density of lead is 10% greater. In general, the authors provide a link to the authoritative work of Dr. De Marcillac, dedicated to the discovery of the radioactive isotope of bismuth.
Response: We do not discuss the “metal density” at all. Evidently, the Reviewer meant the term “heavy metal” used several times in the manuscript. The term is not about “density”, the metals in the lower part of the Periodic Table are commonly called “heavy metals”.
As for the use of bismuth compounds for the treatment of oncological diseases, the authors are disingenuous and deliberately mislead readers. In fact, there are experimental studies showing the effectiveness of bismuth compounds against certain forms of cancer cells. There is no real bismuth drug approved by the Ministry of Health of any country for the treatment of cancer and there probably won’t be for a long time. Probably, the authors need to slightly adjust the categoricalness of their statements.
Response: The Authors clearly described the anticancer properties of both bismuth compounds and bismuth nanformulations, paying particular attention to their antileukemic activity and providing necessary references.
2. Materials and methods. The authors write: “Ultra-high-purity freshly deionized water (ρ ≥ 18.0 MΩcm)…”. Judging by the resistivity, it's just reverse osmosis water. Please correct the categorical statements.
Response: Ultra-high-purity (SILAR-grade) freshly deionized water was used in the present study for all chemical experiments. That means water obtained by further ultra-purification of a “common” deionized water obtained by reverse osmosis. Therefore, the term was used correctly.
Section 2.4.3 Cytotoxicity assays - there is probably no point in describing generally accepted methods in detail; you can simply refer to the original source. All researchers are aware of how to use these methods.
Response: Possibly, some researchers more familiar with the methodology may consider the description unnecessary. However, broader readership of the journal as well as some other Reviewers of the present manuscript have different opinion and support the detailed description of the methods, even of those generally accepted. Moreover, we made some slights modifications to the standard procedure and decided to provide a detailed description. That will make it easier for other researchers to reproduce our experiment.
3. Dynamic light scattering. DLS, without major modifications, in principle cannot determine hundredths of nanometers, as for example stated in Table 1 (69.91 nm or 53.76 nm). The authors need to describe what approach allowed them to achieve such amazing accuracy. What the authors changed in the optical design of the device and the mathematics that calculates the autocorrelation function.
Reponse: The values obtained by DLS were rounded to integers.
4. The marks in Figure 1 are not readable. Please fix it!
Response: Indeed, the microimages produced by TEM contain a scale bar, which is generated by the instrument. It could be seen without manipulation of the figure. The original images are provided with the revised version of the manuscript.
5. The labels of the axes of Figure 2 are not readable. Please fix it! Without understanding what is shown in the figure, it is very difficult to review a manuscript.
Response: Corrected
6. Figure 3 presents data on the rate of infection of mouse peritoneal macrophages infected with L. (L.) amazonensis and L. (V.) guyanensis. Looking at the data, we can say that there are no effects! To argue otherwise, the authors should conduct a statistical study and present the data as means plus or minus the standard error of the mean. Place the appropriate stars.
Response: Presentation of the Figure 3 was improved.
7. Comparison of leukemic cells and monkey epithelial cells is not correct in essence. Authors should provide an explanation in the manuscript why this particular model was chosen.
Response: In the assessment of cytotoxicity, Vero cell lines were utilized to evaluate the safety profile of bismuth nanoparticles. Vero cells, which are non-cancerous kidney epithelial cells derived from African green monkeys, are routinely employed in toxicological studies due to their standard morphology and stable growth characteristics. These cells provide a consistent model for examining the safety of various pharmaceuticals, including vaccines, and are particularly useful for their high sensitivity to toxins and ability to mimic human cellular responses. Thus, the choice of Vero cells in this context is aligned with their widespread acceptance in cytotoxicity testing, providing a relevant and reliable system for preliminary safety evaluation of bismuth nanoparticles.
8. The discussion is quite massive and is written, by and large, clearly, although not always appropriately. I do not understand why the authors do not build on their success in the discussion and do not try to extend their discovery to other areas. It can be assumed that the nanoparticle preparations developed by the authors will be in demand for protecting the surfaces of food production, developing antibacterial coatings, packaging, and other things. As an example, I advise you to read Chapter 8 of Article 10.3367/UFNe.2023.09.039577.
Response: The consortium had clearly formulated objectives during the realization of the present research. We may consider the suggested areas in future studies.
Reviewer 3 Report
Comments and Suggestions for Authors
The search for new therapies for both leishmaniasis and leukemia are very timely and important, and it is of interest to explore the use of bismuth in these areas. This manuscript also indicates the use of nanoparticles in containing bismuth as a new therapeutic direction. However, this manuscript is not ready for publication for a number of reasons as detained below:
1. The title does not indicate the work with leukemia cells and thus should be modified as well as the key words section.
2. The materials and methods section requires much more information to allow the reader to really understand this work as detained below:
a. Line 38: indicate for which cell types the BiNPs were not toxic.
b. Line 127: use IUPAC nomenclature for compounds
c. Line 130: what is pore size for cellulose membrane
d. Lines 144-145: it is unclear what the final pH of this preparation is (and I assume it is fairly acidic) and was this the preparation used in the studies with cells or was it diluted into other solvent(s). More information is needed here.
3. Lllllline163 and other places: how were cells counted?
4. Line 176: what solvent used to make the various BiNP solutions?
5. Line 179: how were ‘viable’ promastigotes determined?
6. Line 194: is this the correct reference?
7. The authors do not provide any references for section 2.4.3.1 (resazurin assay)
8. In section 2.4.3.2 9as well as other places in the manuscript, the authors here now report additions as ug/mL rather than nM as used prior in the manuscript. It is more useful to use the same units.
9. In section 2.4.4: What criteria were used to select colonies (size??) more information is needed here.
10. For Table 1: are these single values or mean values for several replicates?
11. Figure 2 is very confusing especially since the axes labels are not consistent. Are these mean ÷ SD values or??? Are the values direct cell viability or percent of some other value?
12. For figure 3, how as the values on the y axis calculated ? Avoid commas in numerals
13. For figure 4, why is DMSO given on the x axis ?
14. Line 323: this should be labeled as figure 5
15. Line 344: this equation is questionable since the authors indicate high molar excel of water thus formation of HCl is highly unlikely; more likely is [H+] and [Cl-] as solvated ions.
16. Lines 346-348 are confusing : Bi-S bonds stable or labile? Please clarify
17. Lines 369-370: what data used to make this statement?
18. Line 374: Figure 3A shows about 80%...please clarify
19. Line407-408: which data were used to support this statement?
Other issues involve some grammar and syntax issues throughout the manuscript; many sentences are too long such as lines 27-29 and 31-34 as examples. In addition, the reference section is incomplete especially references 35 and 36.
Comments on the Quality of English LanguageOther issues involve some grammar and syntax issues throughout the manuscript; many sentences are too long such as lines 27-29 and 31-34 as examples. In addition, the reference section is incomplete especially references 35 and 36.
Author Response
The Authors would like to thank the Reviewer for his/her time dedicated to evaluation of our manuscript.
The search for new therapies for both leishmaniasis and leukemia are very timely and important, and it is of interest to explore the use of bismuth in these areas. This manuscript also indicates the use of nanoparticles in containing bismuth as a new therapeutic direction. However, this manuscript is not ready for publication for a number of reasons as detained below:
1. The title does not indicate the work with leukemia cells and thus should be modified as well as the key words section.
Response: Done
2. The materials and methods section requires much more information to allow the reader to really understand this work as detained below:
a. Line 38: indicate for which cell types the BiNPs were not toxic.
Response: Done
b. Line 127: use IUPAC nomenclature for compounds
Response: Done
c. Line 130: what is pore size for cellulose membrane
Response: according to the manufacturer’s data (https://www.sigmaaldrich.com/FI/en/product/sigma/d9777), the dialysis tubing will retain most proteins of molecular weight 12,000 or greater. That kind of membrane is suitable for successful purification of the nanoparticles obtained.
d. Lines 144-145: it is unclear what the final pH of this preparation is (and I assume it is fairly acidic) and was this the preparation used in the studies with cells or was it diluted into other solvent(s). More information is needed here.
Response: The paragraph quoted by the Reviewer relates to the description of Bi quantification by AES. Indeed, the medium is highly acidic. However, that part of the work has no relation to biological activity studies, it is an analytical procedure.
3. Lllllline163 and other places: how were cells counted?
Response: As described in the manuscript, the cells were quantified using a hemocytometer with the aid of trypan blue dye and a microscope.
4. Line 176: what solvent used to make the various BiNP solutions?
Response: As described above (in the manuscript), deionised water was used as a solvent. All necessary cell culture media and supplements were added as described for biological experiments.
5. Line 179: how were ‘viable’ promastigotes determined?
Response: Viable promastigotes were quantified in a hemocytometer, using a trypan blue dye and an optical microscope with 400x magnification.
6. Line 194: is this the correct reference?
Response: The reference is correct.
7. The authors do not provide any references for section 2.4.3.1 (resazurin assay)
Response: The reference was provided.
8. In section 2.4.3.2 9as well as other places in the manuscript, the authors here now report additions as ug/mL rather than nM as used prior in the manuscript. It is more useful to use the same units.
Response: Measurement units of mg/mL or μg/mL are typically used in protocols for in vitro assays. Therefore, those units were used in the appropriate places of the manuscript. Corresponding µM equivalents were added for clarity.
9. In section 2.4.4: What criteria were used to select colonies (size??) more information is needed here.
Response: An explanation of how the cells were counted was provided as requested.
10. For Table 1: are these single values or mean values for several replicates?
Response: The values of DLS analysis are always a mean value of several replicates.
11. Figure 2 is very confusing especially since the axes labels are not consistent. Are these mean ÷ SD values or??? Are the values direct cell viability or percent of some other value?
Response: To avoid a confusion, the Figure 2 was revised.
12. For figure 3, how as the values on the y axis calculated ? Avoid commas in numerals
Response: Corrected.
13. For figure 4, why is DMSO given on the x axis ?
Response: As stated in 2.4.3.2, we used DMSO as the positive control for the assay. DMSO is a well-established choice for cytotoxicity assays, including the MTT assay, because it disrupts cell membranes and reduces cell viability.
14. Line 323: this should be labeled as figure 5
Response: Corrected.
15. Line 344: this equation is questionable since the authors indicate high molar excel of water thus formation of HCl is highly unlikely; more likely is [H+] and [Cl-] as solvated ions.
Response: Corrected.
16. Lines 346-348 are confusing : Bi-S bonds stable or labile? Please clarify
Response: The explanation provided in the lines quoted is clear (please, consider the whole sentence). The Bi-S bond is thermodynamically more stable then Bi-O one; at the same time, Bi-S is kinetically more liable.
17. Lines 369-370: what data used to make this statement?
Response: The data were obtained in the present study.
18. Line 374: Figure 3A shows about 80%...please clarify
Response: The line quoted refers to a discussion of data obtained in ref. [41-42] of the original manuscript ([43-44] of the revised one). Those data are in a good agreement with our results.
19. Line407-408: which data were used to support this statement?
Response: The data supporting this statement are presented in Figures 4 and 5 that illustrate the higher cytotoxicity of bismuth nanoparticles towards cancerous cells compared to non-cancerous cells. This suggests a selective action of the nanoparticles in targeting cancer cells. Our findings are in a good agreement with recent studies indicating the potential of bismuth nanoparticles for different biomedical applications, including cancer therapy [1, 2].
1. The versatile biomedical applications of bismuth-based nanoparticles and composites: therapeutic, diagnostic, biosensing, and regenerative properties. Chemical Society Reviews (RSC Publishing). https://doi.org/10.1039/C9CS00283A.
2. Passemard, S., Staedler, D., Sonego, G. et al. Functionalized bismuth ferrite harmonic nanoparticles for cancer cells labeling and imaging. J Nanopart Res 17, 414 (2015). https://doi.org/10.1007/s11051-015-3218-8
Other issues involve some grammar and syntax issues throughout the manuscript; many sentences are too long such as lines 27-29 and 31-34 as examples. In addition, the reference section is incomplete especially references 35 and 36.
Response: Corrected. The incomplete references appeared due to an error in Mendeley reference management software.
Reviewer 4 Report
Comments and Suggestions for Authors
The manuscript “A Second Wind for Inorganic APIs: Leishmanicidal and Antiproliferative Activity of Hydrated Bismuth Oxide Nanoparticles” discusses the potential of bismuth nanoparticles in leishmanicidal and antiproliferative activity. The authors have synthesized bismuth nanoparticles and evaluated the efficacy by measuring the parasite viability and infectivity rate. The observation here indicates that the developed nanoparticles possess good cytotoxic activity against K562 and HL60 cells. However, there are significant aspects of the manuscript that could be improved to increase confidence in the results of this study.
1. A better description of the preparation of hydrated bismuth oxide nanoparticles is ideal. Just mentioning the author's previous article is not sufficient.
2. How about the results of the bismuth assay? The inclusion of a representative atomic emission spectroscopy image will give a better understanding.
3. Why did the authors not include the electron diffraction X-ray spectrum of prepared nanoparticles?
4. Similarly, a size distribution curve is generally required in such manuscripts.
5. The images of the results of flow cytometry could be more promising.
6. Authors should reduce the plagiarism, especially with the first source https://www.scielo.br/j/aa/a/zGfhXPfmQxVkLNzXy9FTkFf/?format=pdf&lang=en
Comments on the Quality of English LanguageIt is Ok.
Author Response
The manuscript “A Second Wind for Inorganic APIs: Leishmanicidal and Antiproliferative Activity of Hydrated Bismuth Oxide Nanoparticles” discusses the potential of bismuth nanoparticles in leishmanicidal and antiproliferative activity. The authors have synthesized bismuth nanoparticles and evaluated the efficacy by measuring the parasite viability and infectivity rate. The observation here indicates that the developed nanoparticles possess good cytotoxic activity against K562 and HL60 cells. However, there are significant aspects of the manuscript that could be improved to increase confidence in the results of this study.
-
A better description of the preparation of hydrated bismuth oxide nanoparticles is ideal. Just mentioning the author's previous article is not sufficient.
Response: The preparation procedure of hydrated bismuth oxide nanoparticles is a subject of two patents in Brazil. Those are owned by the INPA and are confidential during the embargo period.
-
How about the results of the bismuth assay? The inclusion of a representative atomic emission spectroscopy image will give a better understanding.
Response: Bismuth assay was performed on a 4100 MP-AES spectrophotometer (Agilent) that does not return images of the spectra, if that was meant in the comment. Following the suggestion of other Reviewers, the standard deviation values were added to the Table 1.
3. Why did the authors not include the electron diffraction X-ray spectrum of prepared nanoparticles?
Response: The comment is unclear, whether the Reviewer is looking for the X-ray diffractometry data (a diffractogram, i.e., intensities and scattering angles of the X-rays diffracted from the material are measured with an X-ray analyzer) or for the X-ray spectoscopy data (an X-ray spectrum, i.e., a graph that shows the intensity of radiation at different wavelengths or the response of the atomic/molecular system to X-ray excitation).
We did not provide the XRD (X-ray diffractometry) data since it is impossible to obtain the BiNPs in question in a dry powder form. Once precipitated or dried, the material suffers an irreversible transformation. Therefore, the method is not suitable for characterisation of the BiNP sol.
The X-ray spectroscopy study was out of the scope of our investigation. First, because our preparation (a sol) is not suitable for the techniques realized under high vacuum conditions. Second, the X-ray spectroscopy is widely used in solid state chemistry research, but the techniques are not informative in terms of biological activity.
-
Similarly, a size distribution curve is generally required in such manuscripts.
Response: We do not agree with the Reviewer, since the DLS is a routine method that produces results on size and ζ-potential distributions reflected by the corresponding values with standard deviations shown in the Table. Even if the journal is in the online format only, there is no necessity to occupy nearly a page of valuable space with figures that are poorly informative (especially, when all necessary information was already provided in the Table). We provide the example of those curves for your information here below.
Figure 1. Size distribution and zeta potential of Bi-MES nanoparticles.
5. The images of the results of flow cytometry could be more promising.
Response: We agree that cytometry, particularly flow cytometry, could be instrumental in supporting our data and better characterising certain activities. Flow cytometry is a powerful tool for validating biological activity and elucidating mechanisms of action, particularly for anticancer agents. While budgetary limitations precluded its use in the current study, future investigations will incorporate cytometry to characterise immunomodulatory activities, given the therapeutic potential of bismuth nanoparticles. It is important to note that the methods employed by us to verify the biological activities were appropriate and recommended for that kind of research.
6. Authors should reduce the plagiarism, especially with the first source https://www.scielo.br/j/aa/a/zGfhXPfmQxVkLNzXy9FTkFf/?format=pdf&lang=en
Response: First, we would like to draw the Reviewer’s attention to appropriate use of the language.
The paper referred to was produced by the same group at the INPA (Brazil) that makes part of the present consortium. Descriptions of the same standard methodologies and discussion patterns used by the same persons may indeed appear very similar, and this is absolutely normal.
Round 2
Reviewer 2 Report
Comments and Suggestions for Authors
The manuscript has been improved to the conditionally satisfactory threshold. The authors use a rather unceremonious style in their correspondence. The device used by the authors cannot measure the electro-kinetic potential with an accuracy of hundredths of millivolts. Mathematics has taught us since school that the mean and the standard error of the mean cannot differ in the number of digits. Please correct. After this, there is no need to send for re-review.
Reviewer 3 Report
Comments and Suggestions for Authors
I thank the authors for their careful and thoughtful revisions